# Take A Shortcut Back: Mitigating the Gradient Vanishing for Training Spiking Neural Networks

**Yufei Guo**[*], **Yuanpei Chen**[*], **Zecheng Hao, Weihang Peng, Zhou Jie, Yuhan Zhang, Xiaode Liu, Zhe Ma**[†]

Intelligent Science & Technology Academy of CASIC, China
School of Computer Science, Peking University, China
yfguo@pku.edu.cn, rop477@163.com, haozecheng@pku.edu.cn, mazhe_thu@163.com

## Abstract

The Spiking Neural Network (SNN) is a biologically inspired neural network infrastructure that has recently garnered significant attention. It utilizes binary spike activations to transmit information, thereby replacing multiplications with additions and resulting in high energy efficiency. However, training an SNN directly poses a challenge due to the undefined gradient of the firing spike process. Although prior works have employed various surrogate gradient training methods that use an alternative function to replace the firing process during back-propagation, these approaches ignore an intrinsic problem: gradient vanishing. To address this issue, we propose a shortcut back-propagation method in the paper, which advocates for transmitting the gradient directly from the loss to the shallow layers. This enables us to present the gradient to the shallow layers directly, thereby significantly mitigating the gradient vanishing problem. Additionally, this method does not introduce any burden during the inference phase. To strike a balance between final accuracy and ease of training, we also propose an evolutionary training framework and implement it by inducing a balance coefficient that dynamically changes with the training epoch, which further improves the network's performance. Extensive experiments conducted over popular datasets using several popular network structures reveal that our method consistently outperforms state-of-the-art methods.

## 1 Introduction

The Spiking Neural Network (SNN) has become a popular neural network due to its efficiency and has been widely used in various fields such as object recognition Li et al. (2021a); Xiao et al. (2021), object detection Kim et al. (2019); Qu et al. (2023), and pose tracking Zou et al. (2023). The SNN operates by using binary spike signals to transmit information. When the membrane potential exceeds the threshold, the spiking neuron fires a spike represented by 1; otherwise, there is no spike represented by 0. This unique information processing paradigm is energy-efficient since it replaces multiplications of weights and activations with simple additions. Additionally, this information processing paradigm can be implemented in an efficient event-driven-based computation manner on neuromorphic hardware Ma et al. (2017); Akopyan et al. (2015); Davies et al. (2018); Pei et al. (2019); Guo et al. (2023a), where the computational unit activates only when a spike occurs. This feature saves energy since the computational unit remains silent when there is no spike. Studies have shown that an SNN can save orders of magnitude more energy than its Artificial Neural Network (ANN) counterpart Akopyan et al. (2015); Davies et al. (2018).

---

[*]Equal Contributions.

[†]Corresponding author.

38th Conference on Neural Information Processing Systems (NeurIPS 2024).

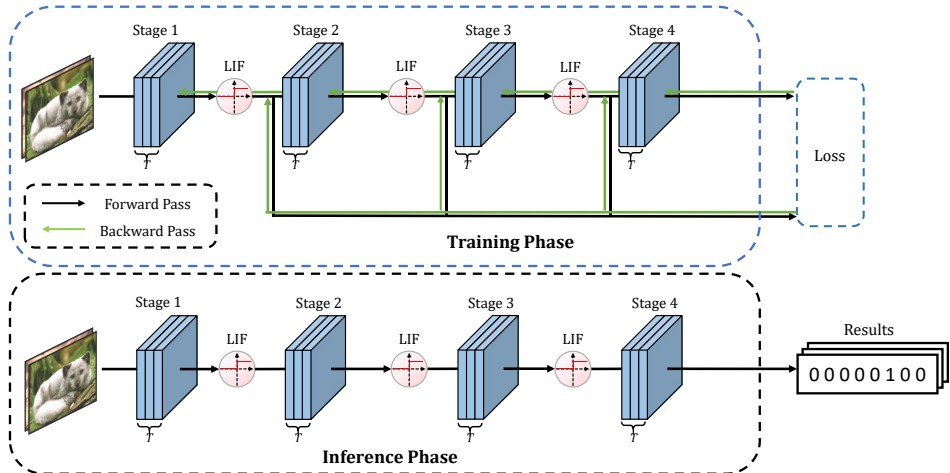

Figure 1: The overall workflow of the proposed method. We add multiple shortcut branches from the intermediate layers to the output thus the gradient from the output could be present to the shallow layers directly.

Although the SNN is energy-efficient, it is challenging to train it directly because the gradient of the firing spike process is not well-defined. This means that it is impossible to use gradient-based optimization methods to train an SNN directly. To overcome this problem, researchers have proposed various surrogate gradient training (SG) methods (Courbariaux et al., 2016; Esser et al., 2016; Bellec et al., 2018; Rathi & Roy, 2020; Wu et al., 2019; Neftci et al., 2019). These methods use an alternative function to replace the firing process during back-propagation. For example, in (Bohte, 2011), (Zenke & Ganguli, 2018), (Guo et al., 2022a), and (Cheng et al., 2020; Guo et al., 2024; Zhang et al., 2024), researchers used the truncated quadratic function, the sigmoid function, the tanh-like function, and the rectangular function as surrogates, respectively. However, SG methods have an intrinsic problem: gradient vanishing. All surrogate functions are bounded, and their gradients would be close to 0 in most intervals. As a result, the gradient of the SNN would quickly decrease from output to input, causing the weights of the shallow layers of the SNN to freeze during optimization. In subsection 4.1, we will theoretically and experimentally demonstrate the gradient vanishing problem.

To address this problem, we propose a shortcut back-propagation method in this paper, which involves transmitting the gradient from the loss to the shallow layers directly. To achieve this, we add multiple shortcut branches from intermediate layers to the output in the network. This allows information from the shallow layers to reach the output and final loss directly. Consequently, the gradient from the output can be present in the shallow layers, and their weights can be updated adequately, resulting in improved accuracy. Importantly, these shortcut branches can be removed without introducing any burden during the inference phase. Our proposed training framework is essentially a joint optimization problem on the weighted sum of the loss functions associated with these shortcut branches. However, if we give more weight to the main branch net, the earlier layer weights may not be updated sufficiently. Conversely, if we give more attention to the side shortcut branches, the accuracy cannot reach a high level since it directly relates to the main branch outputs rather than the side branch outputs. To balance this conflict, we introduce an evolutionary training framework. During early training, we pay more attention to the side branch net, allowing sufficient weight updates of the shallow layers. Towards the end of training, we increase the weight given to the main branch net, which further improves final accuracy. We accomplish this by inducing a dynamically changing balanced coefficient that adjusts with each training epoch. To illustrate our method's workflow, please refer to Figure 1. Our paper provides several key contributions, which can be summarized as follows:

- Firstly, we have identified that the gradient vanishing problem is a significant issue for SNNs. We have supported this conclusion with theoretical justifications and in-depth experimental analysis. To mitigate this problem, we have proposed the shortcut back-propagation approach, which is a simple yet effective method. Importantly, it will not introduce any additional burden during the inference phase.

- Secondly, we have proposed an evolutionary training framework that balances the weights of these branches with a gradual strategy. This approach ensures that earlier layer weights can be adequately updated while also improving overall accuracy.

- Lastly, we have evaluated the effectiveness and efficiency of our proposed methods on both static (CIFAR-10, CIFAR-100, ImageNet) and spiking (CIFAR10-DVS) datasets with widely used backbones. Our results demonstrate that the SNN trained with our proposed method is highly effective, achieving a top-1 accuracy of 77.79% on CIFAR-100 using ResNet19 with only 2 timesteps. This represents a significant improvement of about 3.3% compared with the current state-of-the-art SNN models even with more timesteps.

## 2 Related Work

### 2.1 Learning of Spiking Neural Networks

Unsupervised learning Diehl & Cook (2015); Hao et al. (2020), converting ANN to SNN (ANN2SNN) Sengupta et al. (2019); Hao et al. (2023a,b), and supervised learning Li et al. (2021b); Guo et al. (2022c) are three commonly used learning paradigms for SNNs. In unsupervised learning, the spike-timing-dependent plasticity (STDP) approach Lobov et al. (2020) is utilized to update the SNN model, which is considered more biologically plausible. However, due to the lack of explicit task guidance, this method is typically limited to small-scale networks and datasets. The ANN-SNN conversion method Han & Roy (2020); Li et al. (2021a); Bu et al. (2021); Ho & Chang (2021); Bu et al. (2022); Hao et al. (2023a); Lan et al. (2023) involves training an ANN first and then converting it into a homogeneous SNN by transferring the trained weights and replacing the ReLU neuron with a temporal spiking neuron. However, this method is not suitable for neuromorphic datasets as the ReLU neuron in the ANN cannot capture the rich temporal dynamics required for sequential information. Supervised learning Fang et al. (2021a); Wu et al. (2018), on the other hand, adopts an alternative function during back-propagation to replace the firing process, enabling direct training of the SNN as an ANN. This approach leverages the success of gradient-based optimization and can achieve good performance with only a few time steps, even on large-scale datasets. Moreover, supervised learning can handle temporal data effectively, making it an increasingly popular choice in SNN research. Our work also falls within this domain.

### 2.2 Relieving Training Difficulties for Supervised Learning of SNNs

As mentioned earlier, the surrogate gradient (SG) approach is commonly employed to address the non-differentiability of SNNs. Various SG functions have been utilized, including the truncated quadratic function (Bohte, 2011), the `sigmoid` function (Zenke & Ganguli, 2018), the `tanh`-like function (Guo et al., 2022a), and the rectangular function (Cheng et al., 2020). While the SG method is generally effective, it can also introduce certain issues. Firstly, there is a gradient mismatch between the true gradient and the surrogate gradient, resulting in slow convergence and reduced accuracy. To tackle this problem, IM-Loss (Guo et al., 2022a) proposed a dynamic manual SG method that adapts with each epoch, ensuring sufficient weight updates and accurate gradients simultaneously. In contrast to this manual design, the Differentiable Spike method (Li et al., 2021b) and the differentiable SG search method determine the optimal gradient estimation using finite difference and NAS techniques, respectively. Secondly, due to the firing function being bounded, all these SG functions are also bounded. As a result, the gradient approaches or reaches close to zero in most intervals, exacerbating the vanishing gradient problem in SNNs. To mitigate this issue, SEW-ResNet (Fang et al., 2021a) suggested using the ResNet with activation before addition form, while MS-ResNet (Hu et al., 2021) advocated for the ResNet with pre-activation form. Additionally, normalization techniques have been employed to address the vanishing/explosion gradient problems. For example, Threshold-dependent batch normalization (tdBN) (Zheng et al., 2021) normalized the data along both the channel and temporal dimensions. Other techniques such as Temporal Batch Normalization Through Time (BNTT) (Kim & Panda, 2021), postsynaptic potential normalization (PSP-BN) (Ikegawa et al., 2022), and temporal effective batch normalization (TEBN) (Duan et al., 2022) recognized the significant variation in spike distributions across different timesteps, and thus regulated spike flows by applying separate timestep batch normalization. MPBN (Guo et al., 2023b) introduced an additional batch normalization step after the membrane potential updating function to reestablish data flow. Similarly, regularization loss has been utilized to alleviate gradient explosion/vanishing problems. In RecDis-

SNN (Guo et al., 2022c), a membrane potential regularization loss was proposed to control spike flow within an appropriate range. In Spiking PointNet (Ren et al., 2023), a trained-less but learning-more paradigm was proposed. This method can be seen as using a small network in the training to mitigate the training difficulty problem.

However, all these methods still need to present the gradient from the output layer to the first layer step by step, thus the gradient vanishing problem cannot be solved completely. In this paper, we propose a shortcut back-propagation method. Different from the above methods, we present the gradient from the output layer to these shallow layers directly, thus the gradient vanishing problem can be solved totally.

## 3 Preliminary

The spiking neuron serves as the fundamental and specialized computing unit in SNNs, drawing inspiration from the human brain. In our paper, we employ the widely used spiking Leaky-Integrate-and-Fire (LIF) neuron model. This model accurately captures the behavior of biological neurons by considering the interaction between the membrane potential and input current. To show the spiking neuron in detail, we introduce the notation first. Throughout the paper, we denote the vectors in bold italic letters. For instance, we use the $x$ and $o$ to represent the input and target output variables. We denote the matrices or tensors by bold capital letters (e.g., $\mathbf{W}$ is for weights). We denote the constants by small upright or downright letters. For example, $u_i^{(t)}$ means the $i$-th membrane potential at time step $t$. Then, the LIF neuron can be described as follows:

$$u^{(t+1),\text{pre}} = \tau u^{(t)} + c^{(t+1)}, \text{where } c^{(t+1)} = \mathbf{W} x^{(t+1)}, \tag{1}$$

where $\tau$ is a constant within $(0, 1)$, which controls the leakage of membrane potential. When $\tau$ is 1, the neuron will degenerate to the Integrate-and-Fire (IF) neuron model. In the paper, we set $\tau$ as 0.5. $u^{(t+1),\text{pre}}$ is the pre-synaptic input at time step $t + 1$, which is charged by the input current $c^{(t+1)}$. Note that we omit the layer index for simplicity. The input current is computed by the dot-product between the weights, $\mathbf{W}$ of the current layer and the spike output, $x^{(t+1)}$ from the previous layer. Once the membrane potential, $u^{(t+1),\text{pre}}$ exceeds the firing threshold $V_{\text{th}}$, a spike will be fired from the LIF neuron, given by

$$o^{(t+1)} = \begin{cases} 1 & \text{if } u^{(t+1),\text{pre}} > V_{\text{th}} \\ 0 & \text{otherwise} \end{cases}, \tag{2}$$
$$u^{(t+1)} = u^{(t+1),\text{pre}} \cdot (1 - o^{(t+1)}).$$

After firing, the spike output $o^{(t+1)}$ will propagate to the next layer and become the input $x^{(t+1)}$ of the next layer. In the paper, we set $V_{\text{th}}$ as 1.

There is a notorious problem in SNN training the firing function is undifferentiable. To demonstrate this problem, we formulate the gradient by the chain rule, given as

$$\frac{\partial L}{\partial \mathbf{W}} = \sum_t \left( \frac{\partial L}{\partial o^{(t)}} \frac{\partial o^{(t)}}{\partial u^{(t),\text{pre}}} + \frac{\partial L}{\partial u^{(t+1),\text{pre}}} \frac{\partial u^{(t+1),\text{pre}}}{\partial u^{(t),\text{pre}}} \right) \frac{\partial u^{(t),\text{pre}}}{\partial \mathbf{W}}. \tag{3}$$

Since the firing function Equation 2 is similar to the $\text{sign}$ function. The $\frac{\partial o^{(t)}}{\partial u^{(t),\text{pre}}}$ is 0 almost everywhere except for the threshold. Therefore, the updates for weights would either be 0 or infinity if we use the actual gradient of the firing function.

## 4 Methodology

### 4.1 The Gradient Vanishing Problem for SNNs

As aforementioned, the non-differentiability of SNNs poses challenges when training them directly. To address this issue, researchers have proposed the use of surrogate gradients. In this approach, the firing function remains unchanged during the forward pass, but a surrogate function is employed during the backward pass. The surrogate gradient is then computed based on this surrogate function.

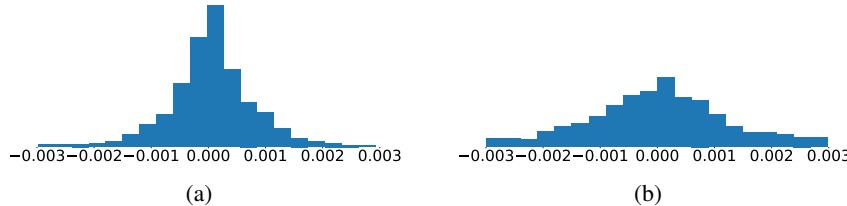

Figure 2: The gradient distributions of the first layer for Spiking ResNet34 on CIFAR-100. (a) and (b) show the distributions for the vanilla model and the one with the shortcut back-propagation method.

There are three commonly used surrogate gradients:

$$
\begin{cases}
\frac{\partial \boldsymbol{o}^{(t)}}{\partial \boldsymbol{u}^{(t),\mathrm{pre}}} &= \gamma \max \left(0, 1 - \left|\frac{\boldsymbol{u}^{(t),\mathrm{pre}}}{V_{\mathrm{th}}} - 1\right|\right), \\
\frac{\partial \boldsymbol{o}^{(t)}}{\partial \boldsymbol{u}^{(t),\mathrm{pre}}} &= \frac{1}{a}\mathrm{sign}\left(\left|\boldsymbol{u}^{(t),\mathrm{pre}} - V_{\mathrm{th}}\right| < \frac{a}{2}\right), \\
\frac{\partial \boldsymbol{o}^{(t)}}{\partial \boldsymbol{u}^{(t),\mathrm{pre}}} &= k(1 - \tanh(\boldsymbol{u}^{(t),\mathrm{pre}} - V_{\mathrm{th}}))^2.
\end{cases}
\tag{4}
$$

Each of these surrogate gradients includes a hyperparameter that controls the sharpness and width of the gradient. However, it is evident that these gradients, or their approximations, often become close to zero over a significant portion of their intervals. Consequently, this poses a considerable challenge in terms of severe gradient vanishing. While residual blocks have proven effective in mitigating gradient vanishing problems in traditional neural networks, their performance is not optimal when applied to SNNs. To demonstrate this, we express the skip connection using the following formulation:

$$
\boldsymbol{o} = g(f(\boldsymbol{x}) + \boldsymbol{x}),
\tag{5}
$$

where $f(\cdot)$ is convolutional layers and $g(\cdot)$ is the activation function. The standard ResNet network is composed of multiple skip connection blocks cascaded together. In ANN, ReLU is used for $g(\cdot)$, since ReLU is unbounded for the positive part, the gradient can be passed to the input of the block directly. However, in the case of LIF neurons in SNNs, the gradient will be reduced through the surrogate gradient. Thus, the standard skip connections still suffer the gradient vanishing problem in SNNs. To visually illustrate this problem, we show the gradient distributions of the first layer for Spiking ResNet34 with 4 timesteps on the CIFAR-100 in the Figure 2(a). It can be seen that these gradients are close to 0 in most intervals, meaning the gradient vanishing problem is very significant for these shallow layers.

### 4.2 The Shortcut Back-propagation Method

Both theoretical analysis and experiments reveal the severity of the gradient vanishing problem in shallow layers of SNNs. In this paper, we propose a shortcut back-propagation method to address this issue. Specifically, the network is divided into several blocks, and we add multiple shortcut branches directly from these blocks to the output, as shown in Figure 1. These blocks are then followed by a global average pooling layer and a fully connected layer, resulting in the final output:

$$
\boldsymbol{o}_{\mathrm{final}} = \sum_{l} b_l(\boldsymbol{x}),
\tag{6}
$$

where the $b_l(\boldsymbol{x})$ represents the output of the $l$-th block. While the original final output can be expressed as

$$
\boldsymbol{o}_{\mathrm{final}} = b_n(\boldsymbol{x}), \quad \text{where } b_l(\boldsymbol{x}) = f_l(b_{l-1}(\boldsymbol{x})).
\tag{7}
$$

In the above equation, the $n$ is the total number of the blocks and $f_l(\cdot)$ is the network of the $l$-th block. To demonstrate how our method alleviates the gradient vanishing problem, let's consider the gradient of the weight in the first layer as an example. For the original case, it can be expressed as

$$
\frac{\partial L}{\partial \mathbf{W}_1} = \frac{\partial L}{\partial b_n(\boldsymbol{x})} \frac{\partial b_n(\boldsymbol{x})}{\partial b_{n-1}(\boldsymbol{x})} \cdots \frac{\partial b_{l+1}(\boldsymbol{x})}{\partial b_l(\boldsymbol{x})} \frac{\partial b_l(\boldsymbol{x})}{\partial \mathbf{W}_1}.
\tag{8}
$$

**Algorithm 1** Training and inference procedure of SNN with our method.

---

Training

**Input**: An SNN to be trained; Initial balance coefficient $\lambda$; training dataset; total training iteration: $I_{\text{train}}$.

**Output**: The well-trained SNN.

1: **for** all $i = 1, 2, \ldots, I_{\text{train}}$ iteration **do**
2:     Get mini-batch training data, $\boldsymbol{x}_{\text{in}}(i)$ and class label, $\boldsymbol{y}(i)$;
3:     Feed the $\boldsymbol{x}_{\text{in}}(i)$ into the SNN and calculate every block output $b_l(\boldsymbol{x}_{\text{in}}(i))$ and the final main net output $b_n(\boldsymbol{x}_{\text{in}}(i))$;
4:     Update $\lambda$;
5:     Calculate the final output by Equation 10;
6:     Compute classification loss $L_{\text{CE}} = \mathcal{L}_{\text{CE}}(\boldsymbol{o}_{\text{final}}(i), \boldsymbol{y}(i))$;
7:     Calculate the gradient w.r.t. $\mathbf{W}$ by Equation 9;
8:     Update $\mathbf{W}$: ($\mathbf{W} \leftarrow \mathbf{W} - \eta \frac{\partial L}{\partial \mathbf{W}}$) where $\eta$ is learning rate.
9: **end for**

Inference

**Input**: The trained SNN; test dataset; total test iteration: $I_{\text{test}}$.

**Output**: The result.

1: **for** all $i = 1, 2, \ldots, I_{\text{test}}$ iteration **do**
2:     Get mini-batch test data, $\boldsymbol{x}_{\text{in}}(i)$ and class label, $\boldsymbol{y}(i)$ in test dataset;
3:     Feed the $\boldsymbol{x}_{\text{in}}(i)$ into the trained SNN;
4:     Calculate the final main net output $b_n(\boldsymbol{x}_{\text{in}}(i))$;
5:     Calculate the final output, $\boldsymbol{o}_{\text{final}}(i) = b_n(\boldsymbol{x}_{\text{in}}(i))$ ;
6:     Compare the final output $\boldsymbol{o}_{\text{final}}(i)$ and $\boldsymbol{y}(i)$ to compute the classification result.
7: **end for**

---

Since the $\frac{\partial b_{l+1}(\boldsymbol{x})}{\partial b_l(\boldsymbol{x})}$ is always reduced through the surrogate gradient, the $\frac{\partial L}{\partial \mathbf{W}_1}$ becomes very small, resulting in insufficient weight updates. However, for our method, it can be expressed as

$$\frac{\partial L}{\partial \mathbf{W}_1} = \sum_l \frac{\partial L}{\partial b_l(\boldsymbol{x})} \frac{\partial b_l(\boldsymbol{x})}{\partial \mathbf{W}_1}. \tag{9}$$

In our method, the gradient is directly fed into the first block and subsequently to $\mathbf{W}_1$. This completely solves the gradient vanishing problem. To further illustrate this advantage, we visualize the gradient distribution of the first layer for Spiking ResNet34 on CIFAR-100 in Figure 2(b). It can be observed that the distribution is relatively flat, indicating that the gradient vanishing problem has been effectively addressed in these shallow layers. Moreover, these shortcut branches can be removed during inference, incurring no additional cost.

### 4.3 The Evolutionary Training Framework

While the shortcut back-propagation method effectively addresses the gradient vanishing problem, it introduces a potential conflict. Each shortcut branch contributes to the final output, but if we focus too much on the shallow layer outputs, the overall accuracy may suffer. This is because the shortcut branches are removed during the inference phase, and the final accuracy is primarily influenced by the main branch. On the other hand, if we prioritize the main branch output, the final loss may not capture enough information from the shallow layers. Consequently, the weights of these shallow layers may not be updated adequately.

To address this issue, we propose an evolutionary training framework. During the early stages of training, we prioritize the former side branch net, allowing for sufficient weight updates in the shallow layers. As training progresses, we gradually shift our focus to the main branch net until all attention is on the main net at the end of training. To achieve this, we introduce a balance coefficient, denoted by $\lambda(i)$ and adopt a strategy of decreasing to adjust it as follows,

$$\boldsymbol{o}_{\text{final}} = b_n(\boldsymbol{x}) + \lambda(i) \sum_{l=1} b_l(\boldsymbol{x}), \text{ where } \lambda(i) = \lambda(1 - \frac{i}{I}). \tag{10}$$

Table 1: Ablation study for the shortcut back-propagation method.

| Architecture | Method | Time-step | Accuracy |
|---|---|---|---|
| ResNet18 | Vanilla Training | 2 | 71.42% |
| | Shortcut Back-propagation | 2 | **73.68%** |
| | Evolutionary Training | 2 | **74.02%** |
| | Vanilla Training | 4 | 72.22% |
| | Shortcut Back-propagation | 4 | **74.78%** |
| | Evolutionary Training | 4 | **74.83%** |
| ResNet34 | Vanilla Training | 2 | 69.82% |
| | Shortcut Back-propagation | 2 | **74.06%** |
| | Evolutionary Training | 2 | **74.17%** |
| | Vanilla Training | 4 | 69.98% |
| | Shortcut Back-propagation | 4 | **75.67%** |
| | Evolutionary Training | 4 | **75.81%** |

In the equation mentioned above, $I$ represents the total number of training iterations, $i$ denotes the current training iteration, and $\lambda$ is a constant. In our work, we set $\lambda$ to a value of 0.25. The training and inference of our SNN are detailed in Algo. 1.

## 5 Experiment

We conduct extensive experiments on CIFAR-10(100) Krizhevsky et al. (2010), ImageNet Deng et al. (2009), and CIFAR10-DVS Li et al. (2017) to demonstrate the superior performance of our method. The CIFAR-10(100) dataset comprises 50k training images and 10k test images, divided into 10(100) classes, each with $32 \times 32$ pixels. CIFAR10-DVS is a converted dataset derived from CIFAR-10. It consists of 10k images, with 1k images per class, in 10 classes. ImageNet is a significantly larger dataset, containing over 1,250k training images and 50k test images. For these static datasets (CIFAR-10, CIFAR-100, and ImageNet), we applied data normalization to ensure that they have 0 mean and 1 variance. Additionally, to prevent overfitting, we performed random horizontal flipping and cropping on all these datasets. For a fair comparison, AutoAugment Cubuk et al. (2018) was also used for data augmentation following these work Guo et al. (2022b); Li et al. (2021b) on CIFAR-10(100). Regarding the CIFAR10-DVS dataset, we partitioned it into 9k training images and 1k test images, as described in Wu et al. (2019). The training image frames were resized to $48 \times 48$ as in Zheng et al. (2021). Random horizontal flipping and random roll within a range of 5 pixels were also applied during training. For the test images, we simply resized them to $48 \times 48$ without any additional processing, following the approach of Li et al. Li et al. (2021b). We run the model with 8 V100.

### 5.1 Ablation Study

To validate the effectiveness of the proposed shortcut back-propagation method, we initially conducted several ablation experiments on the CIFAR-100 dataset using ResNet18 and ResNet34 with different timesteps. The results are detailed in Table 1. For ResNet18, the accuracy achieved through vanilla training is 71.42% and 72.22% under 2 and 4 timesteps, respectively, which aligns with existing works. Upon applying our shortcut back-propagation method, the accuracy improved to 73.68% and 74.78%, marking a notable 2.5% enhancement. Furthermore, with the evolutionary training method, the performance of ResNet18 saw an additional improvement, reaching 74.02% and 74.83%, respectively. Under vanilla training, ResNet34 achieved accuracies of 69.82% and 69.98% with 2 and 4 timesteps, respectively. These results are actually worse than those obtained with ResNet18. This suggests that the deeper model does not exhibit better performance due to the significant gradient vanishing problem in SNNs. However, by utilizing our shortcut back-propagation method, the accuracy significantly improves to 74.06% and 75.67%, representing a remarkable 5.0% enhancement. Notably, these results surpass the performance of ResNet18 as well. This clearly demonstrates the effectiveness of our proposed method. Furthermore, when incorporating the evolutionary training method, we observe further improvements in performance.

Table 2: Comparison with SoTA methods on CIFAR-10(100).

| Dataset | Method | Type | Architecture | Timestep | Accuracy |
|---|---|---|---|---|---|
| CIFAR-10 | TL Wu et al. (2021a) | Tandem Learning | CIFARNet | 8 | 89.04% |
| | PTL Wu et al. (2021b) | Tandem Learning | VGG11 | 16 | 91.24% |
| | PLIF Fang et al. (2021b) | SNN training | PLIFNet | 8 | 93.50% |
| | DSR Meng et al. (2022) | SNN training | ResNet18 | 20 | 95.40% |
| | KDSNN Xu et al. (2023) | SNN training | ResNet18 | 4 | 93.41% |
| | RecDis-SNN Guo et al. (2022c) | SNN training | ResNet19 | 2 | 93.64% |
| | Diet-SNN Rathi & Roy (2020) | SNN training | ResNet20 | 5 | 91.78% |
| | | | | 10 | 92.54% |
| | Dspike Li et al. (2021b) | SNN training | ResNet20 | 2 | 93.13% |
| | | | | 4 | 93.66% |
| | STBP-tdBN Zheng et al. (2021) | SNN training | ResNet19 | 2 | 92.34% |
| | | | | 4 | 92.92% |
| | TET Deng et al. (2022) | SNN training | ResNet19 | 2 | 94.16% |
| | | | | 4 | 94.44% |
| | Real Spike Guo et al. (2022d) | SNN training | ResNet19 | 2 | 95.31% |
| | | | ResNet20 | 4 | 91.89% |
| | **Shortcut Back-propagation** | SNN training | ResNet18 | 2 | **93.89%**±0.11 |
| | | | | 4 | **94.30%**±0.09 |
| | | | ResNet19 | 1 | **94.47%**±0.09 |
| | | | | 2 | **95.19%**±0.10 |
| | **Evolutionary Training** | SNN training | ResNet18 | 2 | **93.92%**±0.08 |
| | | | | 4 | **94.46%**±0.11 |
| | | | ResNet19 | 1 | **94.81%**±0.13 |
| | | | | 2 | **95.36%**±0.10 |
| CIFAR-100 | T2FSNN Park et al. (2020) | ANN2SNN | VGG16 | 680 | 68.80% |
| | Real Spike Guo et al. (2022d) | SNN training | ResNet20 | 5 | 66.60% |
| | LTL Yang et al. (2022) | Tandem Learning | ResNet20 | 31 | 76.08% |
| | Diet-SNN Rathi & Roy (2020) | SNN training | ResNet20 | 5 | 64.07% |
| | RecDis-SNN Guo et al. (2022c) | SNN training | ResNet19 | 4 | 74.10% |
| | Dspike Li et al. (2021b) | SNN training | ResNet20 | 2 | 71.68% |
| | | | | 4 | 73.35% |
| | TET Deng et al. (2022) | SNN training | ResNet19 | 2 | 72.87% |
| | | | | 4 | 74.47% |
| | **Shortcut Back-propagation** | SNN training | ResNet18 | 2 | **73.68%**±0.10 |
| | | | | 4 | **74.78%**±0.08 |
| | | | ResNet19 | 1 | **75.75%**±0.10 |
| | | | | 2 | **77.56%**±0.13 |
| | **Evolutionary Training** | SNN training | ResNet18 | 2 | **74.02%**±0.09 |
| | | | | 4 | **74.83%**±0.11 |
| | | | ResNet19 | 1 | **75.82%**±0.12 |
| | | | | 2 | **77.79%**±0.08 |

## 5.2 Comparison with SoTA Methods

In this section, we conducted a comparative experiment for the shortcut back-propagation method and the evolutionary training framework, taking into consideration several state-of-the-art works. To ensure a fair comparison, we present the top-1 accuracy results based on 3 independent trials. We first evaluated our method on CIFAR-10 and CIFAR-100 datasets. The AdamW optimizer was employed with a learning rate of 0.01 which is cosine decay to 0 and a weight decay of 0.02. Throughout the training process, all models were trained using a batch size of 128 for a total of 300 epochs. The results are summarized in Table 2. For the CIFAR-10 dataset, we chose SpikeNorm Sengupta et al. (2019), Hybrid-Train Rathi et al. (2020), TSSL-BP Zhang & Li (2020), TL Wu et al. (2021a), PTL Wu et al. (2021b), PLIF Fang et al. (2021b), DSR Meng et al. (2022), KDSNN Xu et al. (2023), Diet-SNN Rathi & Roy (2020), Dspike Li et al. (2021b), STBP-tdBN Zheng et al. (2021), TET Deng et al. (2022), RecDis-SNN Guo et al. (2022c), and Real Spike Guo et al. (2022d) as our comparison. Previous works utilizing ResNet18, ResNet19, and ResNet20 as backbones achieved the highest accuracies of 95.40%, 95.31%, and 93.66% with 20, 2, and 4 timesteps respectively. While our method based on ResNet18 and ResNet19 could reach 93.92% and 95.36% with 4 and 2 timesteps, respectively. Note that, ResNet18 is smaller than ResNet20. On the CIFAR-100 dataset, our method can also achieve better accuracy than other prior state-of-the-art works with fewer timesteps. For instance, our ResNet19 model with only 2 timesteps outperforms the current best method, TET and

Table 3: Comparison with SoTA methods on ImageNet.

| Method | Type | Architecture | Timestep | Accuracy |
|---|---|---|---|---|
| TET Deng et al. (2022) | SNN training | ResNet34 | 6 | 64.79% |
| RecDis-SNN Guo et al. (2022c) | SNN training | ResNet34 | 6 | 67.33% |
| GLIF Yao et al. (2022) | SNN training | ResNet34 | 4 | 67.52% |
| DSR Meng et al. (2022) | SNN training | ResNet18 | 50 | 67.74% |
| MS-ResNet Hu et al. (2023) | SNN training | ResNet18 | 6 | 63.10% |
| MPBN Guo et al. (2023b) | SNN training | ResNet18 | 4 | 63.14% |
| | | ResNet34 | 4 | 64.71% |
| Real Spike Guo et al. (2022d) | SNN training | ResNet18 | 4 | 63.68% |
| | | ResNet34 | 4 | 67.69% |
| SEW ResNet Fang et al. (2021a) | SNN training | ResNet18 | 4 | 63.18% |
| | | ResNet34 | 4 | 67.04% |
| **Shortcut Back-propagation** | SNN training | ResNet18 | 4 | **64.47%**±0.21 |
| | | ResNet34 | 4 | **67.90%**±0.17 |
| **Evolutionary Training** | SNN training | ResNet18 | 4 | **65.12%**±0.18 |
| | | ResNet34 | 4 | **68.14%**±0.15 |

Table 4: Comparison with SoTA methods on CIFAR10-DVS.

| Method | Type | Architecture | Timestep | Accuracy |
|---|---|---|---|---|
| STBP-tdBN Zheng et al. (2021) | SNN training | ResNet19 | 10 | 67.80% |
| RecDis-SNN Guo et al. (2022c) | SNN training | ResNet19 | 10 | 72.42% |
| Real Spike Guo et al. (2022d) | SNN training | ResNet19 | 10 | 72.85% |
| Dspike Li et al. (2021b) | SNN training | ResNet18 | 10 | 75.40% |
| **Shortcut Back-propagation** | SNN training | ResNet18 | 10 | **82.00%**±0.10 |
| **Evolutionary Training** | SNN training | ResNet18 | 10 | **83.30%**±0.10 |

RecDis-SNN even with 4 timesteps by about 3.3%. These experimental results clearly demonstrate the efficiency and effectiveness of our method.

We proceeded to conduct experiments on the ImageNet dataset, which is a more complex dataset than CIFAR. The learning rate is adjusted to $4e^{-3}$ here. The comparative results are presented in Table 3. Notably, there have been several state-of-the-art (SoTA) baselines proposed for this dataset recently, such as RecDis-SNN Guo et al. (2022c), GLIF Yao et al. (2022), DSR Meng et al. (2022), MPBN Guo et al. (2023b), MS-ResNet Hu et al. (2023), Real Spike Guo et al. (2022d), and SEW ResNet Fang et al. (2021a). It is important to note that Real Spike and SEW ResNet deviate from the typical ResNet backbone as they generate integer outputs in the intermediate layers, making them more energy-intensive compared to methods with standard backbones. In contrast, our approach adopts the standard ResNet18 and ResNet34 architectures, yet it still outperforms Real Spike and SEW ResNet. Specifically, our method achieves an accuracy of 65.12% and 68.14% using ResNet18 and ResNet34, respectively, surpassing Real Spike by 1.44% and 0.45%, respectively. This improvement is noteworthy and demonstrates the effectiveness of our method in handling large-scale datasets.

In addition to the aforementioned experiments, we conducted tests on the highly popular neuromorphic dataset, CIFAR10-DVS. We use the same hyper-parameter setting as CIFAR. Employing ResNet18 as the foundational architecture, which is notably smaller compared to ResNet19, our approach achieved remarkable accuracies of 82.00% and 83.30%, respectively. These results demonstrate a substantial improvement over previous methodologies.

## 6 Conclusion

In the paper, we proved that the Spiking Neural Network suffers severe gradient vanishing with theoretical justifications and in-depth experimental analysis. To mitigate the problem, we proposed a shortcut back-propagation method. This enables us to present the gradient to the shallow layers directly, thereby significantly mitigating the gradient vanishing problem. Additionally, this method does not introduce any burden during the inference phase. we also presented an evolutionary training framework by inducing a balance coefficient that dynamically changes with the training epoch, which could further improve the accuracy. We conducted various experiments to verify the effectiveness of our method.

## Acknowledgment

This work is supported by grants from the National Natural Science Foundation of China under contracts No.12202412 and No.12202413.

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
