# OpenReview forum: "Take A Shortcut Back: Mitigating the Gradient Vanishing for Training Spiking Neural Networks"
_NeurIPS.cc/2024/Conference — NeurIPS 2024 poster_

### Official Review · Reviewer_Jb1s · 2024-06-18

**Soundness:** 3
**Presentation:** 3
**Contribution:** 3
**Rating:** 6
**Confidence:** 5

**Summary:**

The paper trains SNNs using surrogate gradient learning. In order to mitigate the gradient vanishing problem, the paper proposed the Shortcut Back-propagation method and utilizes an evolutionary algorithm framework to balance the training of shallow and deep layers. The effectiveness of the proposed method is demonstrated through many experiments.

**Strengths:**

1)	The shortcut backpropagation method and the evolutionary training method are novel.
2)	This paper can well handle the gradient vanishing problem.
3)	The paper is well-written.
4)	The paper shows the effectiveness of the proposed methods through many experiments.

**Weaknesses:**

1)	The author should add more mathematical proof to demonstrate that the mentioned residual structure in SNN is not very effective? The introduction of shortcut branches might add complexity to the network architecture, which could affect the interpretability of the model.
2)	Some recent SOTA works should be compared with too.  The authors can also compare with paper [1][2] which obtains really good results by MS-ResNet-18 backbone with 1 or 6 timesteps on large imageNet datasets.


[1]Yao M, Zhao G, Zhang H, et al. Attention spiking neural networks[J]. IEEE transactions on pattern analysis and machine intelligence, 2023.

[2] Qiu X, Zhu R J, Chou Y, et al. Gated attention coding for training high-performance and efficient spiking neural networks[C]. Proceedings of the AAAI Conference on Artificial Intelligence. 2024, 38(1): 601-610.

**Questions:**

1)	Why are the bolded values not always the best values?

**Limitations:**

I find no limitation about the paper.

---

> ### Author Rebuttal · Authors · 2024-08-06
>
> Thanks for your efforts in reviewing our paper and your recognition of our novel method, notable results, and good writing. The response to your questions is given piece by piece as follows.
>
> W**1**:  The author should add more mathematical proof to demonstrate that the mentioned residual structure in SNN is not very effective? The introduction of shortcut branches might add complexity to the network architecture, which could affect the interpretability of the model.
>
> **A1**: Thanks for this advice. **The** **residual structure cannot solve the gradient vanishing problem of SNNs completely.**  For standard skip connections, it can be expressed as $o=g(f(x)+x)$, where f(x) is convolutional layers and g() is the activation function. The standard ResNet network is composed of multiple skip connection blocks cascaded together. In ANN, ReLU is used for g(), since ReLU is unbounded for the positive part, the gradient can be passed to the input of the block directly. However, in the case of LIF neurons in SNNs, the gradient will be reduced through the surrogate gradient. Thus, the skip connections in standard ANNs still suffer the gradient vanishing problem in SNNs, as Figure 2 in the paper also visually illustrates this problem. However, our shortcut back method can transfer the gradient from the output to the input of the block directly in SNNs.
>
> Furthermore, the proposed methods are only alive in the training phase and will not affect the inference phase of SNN.
>
> ---
>
> W2:  Some recent SOTA works should be compared with too. The authors can also compare with paper [1][2] which obtains really good results by MS-ResNet-18 backbone with 1 or 6 timesteps on large imageNet datasets.
>
> **A2**: Thanks for this advice.  We will add comparisons of these recent methods in the revised version as your suggestion. Thanks.
>
> ---
>
> Q1:  Why are the bolded values not always the best values?
>
> **A1**: Sorry for the confusion.  With the same model and same timesteps, our method reaches the best.  We will further clarify this in the revised version.

---

> > ### Comment · Reviewer_Jb1s · 2024-08-08
> >
> > Thank you for your reply. I think this is a nice bit of discussion and could be added to the manuscript.
> > In light of the additional discussion, I'd like to raise my score to a 6. This is an interesting piece of work and would be a nice addition to NeurIPS.

---

> > > ### Author Response · Authors · 2024-08-11
> > > **thanks**
> > >
> > > Thanks very much for your reply and recognition. We are happy to see that your concerns have been addressed.

---

### Official Review · Reviewer_6ctb · 2024-07-07

**Soundness:** 3
**Presentation:** 3
**Contribution:** 3
**Rating:** 6
**Confidence:** 4

**Summary:**

This paper proposes a simple method to mitigate the gradient vanishing problem in the training of SNNs. This method introduces some early classification heads (including a pooling layer and a fully connected layer) to the SNN. Because the gradients from the early classification heads pass fewer surrogate gradients, this method aids the SNN in addressing the gradient vanishing problem. The authors also suggest an evolutionary training framework that changes the loss function to gradually adjust how important early classification head outputs are during the training phase. The proposed methods are only alive in the training phase and will not affect the inference phase of SNN.

**Strengths:**

This proposed method partially alleviates the gradient vanishing problem in the training of SNN with surrogate gradients. Furthermore, the method has demonstrated excellent performance across multiple datasets. The Short-BP method can be easily integrated into the SNN training process without introducing excessive computational overhead. Furthermore, the evolutionary training framework effectively mitigates the short-BP problem, which may make the network pay more attention to early classification heads than the final SNN output. The writing in this paper is clear and concise.

**Weaknesses:**

1. In this paper, the author only demonstrates a change in gradient distribution in the first layer. Presenting the changes in the men and variance of the absolute gradients for each layer would provide a more direct proof of their argument.
2. The author should provide a more detailed mathematical proof to explain why the use of surrogate gradients in deep SNN would lead to gradient vanishing, as well as why direct use of residual learning will not address the problem.
3. The author has not demonstrated their method on much deeper network architectures where the gradient vanishing problem is more severe.

**Questions:**

1. How is the network divided into multiple blocks? Are there any additional rules for the insertion position and number of early classification heads?
2. The results of using short-BP to train ResNet 18 in Table 1 and Table 2 are quite different. There may be a transcription error here.

---

> ### Author Rebuttal · Authors · 2024-08-06
>
> Thanks for your efforts in reviewing our paper and your recognition of our simple method, notable results, and good writing. The response to your questions is given piece by piece as follows.
>
> W**1**:  In this paper, the author only demonstrates a change in gradient distribution in the first layer. Presenting the changes in the men and variance of the absolute gradients for each layer would provide a more direct proof of their argument.
>
> **A1**: Thanks for this advice. Here we add the changes in the men and variance of the absolute gradients for the first 10 layers in vanilla ResNet18 our ResNet18 respectively on the CIFAR-100. It can be seen that our ResNet18 could handle the gradient vanishing problem well.
>
> | Layer | Vanilla  | Ours |
> | --- | --- | --- |
> | 10 | 0.2094/0.0361 | 0.3846/0.0739 |
> | 9 | 0.1460/0.0158 | 0.2489/0.0432 |
> | 8 | 0.1136/0.0083 | 0.2001/0.0243 |
> | 7 | 0.0957/0.0056 | 0.1699/0.0173 |
> | 6 | 0.0753/0.0034 | 0.1344/0.0106 |
> | 5 | 0.0638/0.0023 | 0.1101/0.0070 |
> | 4 | 0.0513/0.0014 | 0.0889/0.0045 |
> | 3 | 0.0349/0.0007 | 0.0611/0.0021 |
> | 2 | 0.0197/0.0002 | 0.0341/0.0006 |
> | 1 | 0.0178/0.0001 | 0.0310/0.0005 |
>
> ---
>
> W2:  The author should provide a more detailed mathematical proof to explain why the use of surrogate gradients in deep SNN would lead to gradient vanishing, as well as why direct use of residual learning will not address the problem.
>
> **A2**: Sorry for the confusion. **The** **residual structure cannot solve the gradient vanishing problem of SNNs completely.**  For standard skip connections, it can be expressed as $o=g(f(x)+x)$, where f(x) is convolutional layers and g() is the activation function. The standard ResNet network is composed of multiple skip connection blocks cascaded together. In ANN, ReLU is used for g(), since ReLU is unbounded for the positive part, the gradient can be passed to the input of the block directly. However, in the case of LIF neurons in SNNs, the gradient will be reduced through the surrogate gradient. Thus, the skip connections in standard ANNs still suffer the gradient vanishing problem in SNNs, as Figure 2 in the paper also visually illustrates this problem. However, our shortcut back method can transfer the gradient from the output to the input of the block directly in SNNs.
>
> ---
>
> W3:  The author has not demonstrated their method on much deeper network architectures where the gradient vanishing problem is more severe.
>
> **A3**: Thanks for this advice.  We have added more experiments for deeper network architectures. It can be seen that our method also works well with these architectures.
>
> | Model | Accuracy |
> | --- | --- |
> | ResNet18 | 72.22% |
> | Our ResNet18 | 74.78% |
> | ResNet34 | 69.98% |
> | Our ResNet34 | 75.67% |
> | ResNet50 | 65.22% |
> | Our ResNet50 | 76.81% |
> | ResNet101 | 53.71% |
> | Our ResNet101 | 76.88% |
> | ResNet152 | 42.53% |
> | Our ResNet152 | 75.49% |
>
> **Q1**:  How is the network divided into multiple blocks? Are there any additional rules for the insertion position and number of early classification heads?
>
> **W1**: Thanks for the question. These modern architectures are usually partitioned into different stages corresponding to a downsample of the input feature map and an increase of the channel numbers. We add the branchs to the exit of these stages.
>
> ---
>
> **Q2**:  The results of using short-BP to train ResNet 18 in Table 1 and Table 2 are quite different. There may be a transcription error here.
>
> **W2**: Thanks for the advice. We confused the results of the Evolutionary Training and the results of the Shortcut Back-propagation. We will correct it in the revised version. Thank you again.

---

> > ### Comment · Reviewer_6ctb · 2024-08-09
> >
> > Thank you for your response. All of my concerns have been addressed, and I'd like to increase my score.

---

> > > ### Author Response · Authors · 2024-08-11
> > > **thanks**
> > >
> > > Thanks very much for your reply and recognition. We are happy to see that your concerns have been addressed.

---

### Official Review · Reviewer_runr · 2024-07-11

**Soundness:** 4
**Presentation:** 3
**Contribution:** 4
**Rating:** 6
**Confidence:** 5

**Summary:**

This paper proposes shortcut connections between layers to mitigate the gradient vanishing problem in SNNs. Additionally, the authors present a way to phase out the shortcut connections over training so that inference can be done without these additional connections. The experiments show that this method improves training performance in several image classification tasks.

**Strengths:**

1.The idea is small, but interesting and effective enough.

2.The performance improvement over the existing SNN methods is noticeable.

3.The paper is well-written.

**Weaknesses:**

1.The proposed method will increase the training time.

2.In the experimental section, some newer methods should be compared with this method.

3.Figure 2 lacks horizontal and vertical coordinates, and the readability and comprehensibility of the picture need to be improved.

**Questions:**

1.Does the proposed method lead to an increase in the calculation of gradient backpropagation? How much is the increased training time.

**Limitations:**

None.

---

> ### Author Rebuttal · Authors · 2024-08-06
>
> Thanks for your efforts in reviewing our paper and your recognition of our interesting ideas, notable results, and good writing. The response to your questions is given piece by piece as follows.
>
> W**1**:  The proposed method will increase the training time.
>
> **A1**: Thanks for this question. The additional training time is trivial. Here we add the training time cost comparison of Vanilla SNN and our Shortcut Back-propagation SNN for 1 epoch training with batchsize as 128 on CIFAR10 using ResNet20 with a single V100. Only about 10% of training cost is induced.
>
> | Method | Timestep | Time |
> | --- | --- | --- |
> | Vanilla SNN | 2 | 22.17s |
> | Shortcut Back-propagation | 2 | 24.49s(10.46%) |
> | Vanilla SNN | 4 | 43.68s |
> | Shortcut Back-propagation | 4 | 47.67s(9.13%) |
>
> ---
>
> **Q2**: In the experimental section, some newer methods should be compared with this method.
>
> **A2**: Thanks for the advice. We will add more recent methods like these in the below papers in the revised version.
>
> [1]Yao M, Zhao G, Zhang H, et al. Attention spiking neural networks[J]. IEEE transactions on pattern analysis and machine intelligence, 2023.
>
> [2] Qiu X, Zhu R J, Chou Y, et al. Gated attention coding for training high-performance and efficient spiking neural networks[C]. Proceedings of the AAAI Conference on Artificial Intelligence. 2024, 38(1): 601-610.
>
> ---
>
> **Q3**: Figure 2 lacks horizontal and vertical coordinates, and the readability and comprehensibility of the picture need to be improved.
>
> **A3**: Thanks for the advice. We have improved the picture as you suggested. Please see the general response.
>
> ---
>
> **W1**: Does the proposed method lead to an increase in the calculation of gradient backpropagation? Howmuch is the increased training time.
>
> **A1**: Thanks for this question. The additional training time is trivial. Here we add the training time cost comparison of Vanilla SNN and our Shortcut Back-propagation SNN for 1 epoch training with batchsize as 128 on CIFAR10 using ResNet20 with a single V100. Only about 10% of training cost is induced.
>
> | Method | Timestep | Time |
> | --- | --- | --- |
> | Vanilla SNN | 2 | 22.17s |
> | Shortcut Back-propagation | 2 | 24.49s(10.46%) |
> | Vanilla SNN | 4 | 43.68s |
> | Shortcut Back-propagation | 4 | 47.67s(9.13%) |

---

### Author Rebuttal · Authors · 2024-08-06

Thanks for your efforts in reviewing our paper and your recognition of our interesting ideas, notable results, and good writing. Here we provide the revised picture in the PDF.

---

### Decision · Program_Chairs · 2024-09-25

**Decision:**

Accept (poster)

**Comment:**

This paper introduces a new method to address the vanishing gradient problem for training spiking neural networks. The method's performance is demonstrated with many experiments. All reviewers agree on acceptance.